# Importance of *Defluviitalea raffinosedens* for Hydrolytic Biomass Degradation in Co-Culture with *Hungateiclostridium thermocellum*

**DOI:** 10.3390/microorganisms8060915

**Published:** 2020-06-17

**Authors:** Regina Rettenmaier, Martina Schneider, Bernhard Munk, Michael Lebuhn, Sebastian Jünemann, Alexander Sczyrba, Irena Maus, Vladimir Zverlov, Wolfgang Liebl

**Affiliations:** 1Chair of Microbiology, Technical University of Munich, Emil-Ramann-Str. 4, 85354 Freising, Germany; regina.rettenmaier@tum.de (R.R.); martina.schneider@tum.de (M.S.); wliebl@wzw.tum.de (W.L.); 2Department for Quality Assurance and Analytics, Bavarian State Research Center for Agriculture, Lange Point 6, 85354 Freising, Germany; bernhard.munk@lfl.bayern.de (B.M.); michael.lebuhn@lfl.bayern.de (M.L.); 3Center for Biotechnology (CeBiTec), Universitätsstr. 27, 33615 Bielefeld, Germany; jueneman@CeBiTec.Uni-Bielefeld.de; 4Faculty of Technology, Bielefeld University, Universitätsstr. 25, 33615 Bielefeld, Germany; asczyrba@techfak.uni-bielefeld.de; 5Institute of Molecular Genetics, RAS, Kurchatov Sq. 2, 123182 Moscow, Russia

**Keywords:** biogas, metagenomics, whole-genome sequencing, cellulose degradation, carbohydrate active enzymes, metabolic interaction

## Abstract

Bacterial hydrolysis of polysaccharides is an important step for the production of sustainable energy, for example during the conversion of plant biomass to methane-rich biogas. Previously, *Hungateiclostridium thermocellum* was identified as cellulolytic key player in thermophilic biogas microbiomes with a great frequency as an accompanying organism. The aim of this study was to physiologically characterize a recently isolated co-culture of *H. thermocellum* and the saccharolytic bacterium *Defluviitalea raffinosedens* from a laboratory-scale biogas fermenter. The characterization focused on cellulose breakdown by applying the measurement of cellulose hydrolysis, production of metabolites, and the activity of secreted enzymes. Substrate degradation and the production of volatile metabolites was considerably enhanced when both organisms acted synergistically. The metabolic properties of *H. thermocellum* have been studied well in the past. To predict the role of *D. raffinosedens* in this bacterial duet, the genome of *D. raffinosedens* was sequenced for the first time. Concomitantly, to deduce the prevalence of *D. raffinosedens* in anaerobic digestion, taxonomic composition and transcriptional activity of different biogas microbiomes were analyzed in detail. *Defluviitalea* was abundant and metabolically active in reactor operating at highly efficient process conditions, supporting the importance of this organism for the hydrolysis of the raw substrate.

## 1. Introduction

The microbial conversion of lignocellulosic biomass to methane-rich biogas has become an important sustainable generating industry [1,2,3]. The process can be divided into several steps with different bacterial or archaeal organisms actively involved: (i) hydrolysis of the insoluble raw substrate by hydrolytic, saccharolytic, and peptolytic Bacteria; (ii) acidogenesis and acetogenesis by acidogenic and/or acetogenic Bacteria, including syntrophic oxidizers of acetate or other intermediates; and (iii) methanogenesis by methanogenic *Archaea* [4] with a diverse spectrum metabolic pathways for methanogenesis, mainly aceticlastic, hydrogenotrophic, or H_2_ and methylotrophic methanogenesis [5]. 

Integrated *omics* approaches describing biogas microbiology previously indicated the presence a huge fraction of unassignable sequences, suggesting that most of the microorganisms in biogas communities are still unknown [6,7,8,9,10]. This is due to insufficient availability of reference strains and their corresponding genome sequences in public databases. Addressing this issue, Maus et al. [11] isolated and genetically characterized novel cellulolytic, hydrolytic, and acidogenic/acetogenic Bacteria as well as methanogenic *Archaea* originating from different anaerobic digestion communities. This study and earlier work [9,11] showed that (hemi)cellulolytic Bacteria mostly represented a minority within the entire biogas microbiome despite their unquestionably pivotal role in initial biomass breakdown, which can be crucial and rate-limiting for the entire biogas process [4,12]. Therefore, it is postulated that few true cellulolytic Bacteria cooperate intimately and synergistically along with saccharolytic and synthrophic Bacteria [13,14,15]. Such cross-feeding has already been reported, e.g., for cellulolytic Bacteria of the genus *Hungateiclostridium* [16], formerly known as *Clostridium* [17,18,19].

Previous analysis of biogas-producing microbial communities revealed, beside *H. thermocellum* as truly cellulolytic organism, many saccharolytic organisms such as *Defluviitalea* spp. accounting for 90% of the relative abundance in cellulolytic mixed isolates or up to 34% in cellulolytic enrichment cultures derived from thermophilic lab-scale biogas fermenters [20]. To unravel the effect of the *H. thermocellum* cellulolytic activity in co-culture with *D. raffinosedens*, the strain *D. raffinosedens* 249c-K6 was isolated in pure culture and its genomic sequence was characterized in detail. This is the first genome of the species *D. raffinosedens* and the second genome of the genus *Defluviitalea* sequenced so far. Further, results of targeted metagenomic and metatranscriptomic sequence analysis are integrated to examine the role of *D. raffinosedens* in the anaerobic biomass decomposition process.

## 2. Materials and Methods

### 2.1. Data Availability

The genome assembly of *D. raffinosedens* 249c-K6 is available at NCBI under Accession No. WSLF00000000 with the Bioproject Accession No. PRJNA591875. Additionally, the raw sequencing reads of the whole genome sequencing were deposited at NCBI with Accession No. SRR10673227. Further, the almost complete 16S rRNA gene sequence of *D. raffinosedens* 249c-K6, as established via direct, gene-specific PCR and Sanger sequencing, is available with Accession No. MN744427. The full-length 16S rRNA gene sequence of *D. raffinosedens* 249c-K6, assembled from the genome sequence and the gene-specific PCR, is available at NCBI with Accession No. MT350287. The sequence datasets of the 16S rRNA amplicon libraries are available in the ENA repository under the Bioproject ID PRJEB37872.

### 2.2. Reactor Operation and Sampling

In this study, two mesophilic (38 °C) and two thermophilic (50 °C) laboratory-scale biogas fermenters (44 L working volume) fed with maize silage only were investigated at different process conditions. Two reactors were operated in parallel for each temperature. All fermenters were started up with inocula from an industrial-scale biogas plant located in Eschelbach an der Ilm, Germany, digesting a feedstock of 72% maize silage, 21% rye whole-crop silage, 3% chicken manure, and small amounts of cow dung, corn cob mix, and grass silage. Inocula were taken from the secondary reactor (operated at 44 °C) or the main fermenter (operated at 46 °C) for mesophilic or thermophilic fermenter operation, respectively. Detailed analyses of inocula are summarized in Appendix A.

After stable anaerobic digestion and biogas production conditions were reached, the processes were transformed to an acidified (mesophilic) or highly efficient (thermophilic) status. Processes are defined stable if no accumulation of volatile fatty acids (VFAs) was observed, and if the methane yield laid in an expected range of 350–400 L_STP_ CH_4_ × kg_VS_^−1^ for the digestion of maize silage. The process was stable in the thermophilic reactors (t_st) at an OLR of 2.0 kg_VS_ × m^−3^ × day^−1^. The OLR was increased, but VFAs did not accumulate, and the methane yield still ranged 350–400 L_STP_ CH_4_ × kg_VS_^−1^ at 4.0 kg_VS_ × m^−3^ × day^−1^. This status was defined as highly efficient (t_ef). In the mesophilic reactors, the process was stable and efficient (m_st) at an OLR of 2.5 kg_VS_ × m^−3^ × day^−1^. VFAs accumulated here in the following, and the methane yield dropped below 300 L_STP_ CH_4_ × kg_VS_^−1^. This disturbed process was defined as acidified (m_ac). Samples were taken from t_st, t_ef, m_st, and m_ac processes for conventional biochemical and microbial community analyses (see below). Fresh samples were processed without delay in the adjacent laboratory.

### 2.3. Organisms

The consortia GS2.5GR_T4 and GS2_O45 originate from a fermenter operation (55 °C) ahead of t_st. Sampling, enrichment, and preliminary identification of these consortia were described by Rettenmaier et al. [20]. Separation of clonal pure isolates from these cultures was performed as described previously [20] or by standard streaking techniques on agar plates and re-inoculation of single colonies in liquid media. Isolate 250c-K4 (*H. thermocellum*) originates from GS2_O45, while the co-culture 253-K6 (*H. thermocellum* and *D. raffinosedens*) and isolate 249c-K6 (*D. raffinosedens*) originate from GS2.5GR_T4. *H. thermocellum* DSM 1237^T^ = ATCC 27405^T^ was purchased from the German Collection of Microorganisms and Cell Cultures (DSMZ, Braunschweig, Germany).

### 2.4. Growth Conditions Used for D. raffinosedens 249c-K6 and H. thermocellum DSM 1237^T^


During this study, the three different cultivation media used for this study were: (i) tap water with 20% (*v*/*v*) sterilized biogas plant digestate, designated GR20; (ii) GS2 medium [21,22]; and (iii) GS2 supplemented with 2.5% (*v*/*v*) digestate, designated GS2.5GR, as also described previously [20]. Filter paper (FP) (Whatmann No.1), crystalline cellulose powder (MN301; Machery Nagel; Düren, Germany), glucose, or xylose was added at 0.2–0.5% (*w/v*). All experimental steps were performed under anaerobic atmosphere (98% N_2_, 2% H_2_) in an anaerobic chamber (Coy Laboratory Products, Grass Lake, MI, USA). Liquid medium was prepared in anaerobic flasks closed with a butyl-rubber stopper. For solid medium, 1.8% (*w/v*) agar was added before autoclaving. Agar plates were incubated anaerobically in AnaeroJars (Thermo Fisher Scientific, Waltham, MA, USA). Ten milliliters of liquid medium were inoculated 1:100 and 50 mL medium for Anthrone assay, gas chromatography (GC), and the determination of enzymatic activities were inoculated 1:50. All cultures were incubated at 55 °C.

### 2.5. Determination of Cellulose Hydrolyzing Activities by the Anthrone Assay of H. thermocellum Alone or in Co-Culture with D. raffinosedens

Cellulolytic cultures consisting of *H. thermocellum* alone or in co-culture with *D. raffinosedens* were pre-cultivated in 10 mL liquid medium (GS2, GS2.5GR, or GR20) with FP at 55 °C. The incubation time varied between two and eight days depending on the cellulolytic property of the culture. One milliliter of the pre-culture was inoculated in 50 mL liquid medium with 0.2% (*w/v*) MN301 as biological duplicates and incubated at 55 °C. Two to eight days after inoculation, aliquots of 2–4 mL were centrifuged (13,000 rpm, 5 min). Pellets were washed twice in 1 mL deionized water (MQ H_2_O), once in 1 mL 100% ethanol, and dried. For swelling of cellulose, 175 µL 72% (*v*/*v*) H_2_SO_4_ was added and incubated for 1 h at room temperature (RT). Samples were filled up to 1 mL with MQ H_2_O and hydrolyzed for 30 min at 100 °C. For glucose assays, samples representing the glucose standard curve (0.02–0.2 mg·mL^−1^) as well as dilutions of the hydrolyzed cellulose samples (1:4/1:10/1:20/1:40) were prepared with MQ H_2_O in a flat-bottom 96-well plate on ice. Two hundred milliliters of ice-cooled Anthrone 0.2% (*w/v*) in 96% (*v*/*v*) H_2_SO_4_ (both Carl Roth, Karlsruhe, Germany) were added to each well, and, subsequently, the plate was incubated in a water bath at 80 °C for 30 min. Afterwards, the absorption was measured at 620 nm with a SPECTROstarnano (BMG LABTECH, Ortenberg, Germany) or Tecan Infinite M200 PRO (Tecan Group Ltd., Männedorf, Switzerland) photometer.

### 2.6. Determination of Volatile Acids and Alcohols with Gas Chromatography

To detect the amount of volatile acids and alcohols in the sample, GC analysis was performed using the GC-2010 system (Shimadzu, Kyoto, Japan) with a flame ionization detector as described previously [22]. In brief, samples were separated with a Stabilwax-DA capillary column (RESTECK, Bad Homburg, Germany) and nitrogen as gas carrier (50 cm × s^−1^). A temperature gradient (12 min from 70 °C to 260 °C followed by 2 min at 260 °C) was used for more effective separation. Two milliliters of fresh liquid culture was centrifuged (3 min, 13,000 rpm), and the supernatant was stored at −20 °C until measurement. Thawed samples were centrifuged again (20 min, 13,000 rpm) and 100 μL of the supernatant was mixed with 350 μL H_2_O (pH 2.0 with HCl) and 50 μL 0.5% (*v*/*v*) 1-propanol as internal standard (final concentration 0.05% (*v*/*v*)). For identification of peaks and quantification of products, an external standard with methanol, ethanol, isopropyl, butanyl, isobutanyl, butyric acid, acetic acid, and propionic acid (0.5% (*v*/*v*) each) was used.

### 2.7. Enzymatic Digestion of Cellulose Biomass and Determination of Enzymatic Activity

Bacterial cells were removed from fresh liquid cultures via centrifugation (4500 rpm, 30 min, 4 °C). Under continuous stirring, saturated ammonium sulfate solution was added in little droplets to the culture supernatant at 4 °C to a final concentration of 60% (*v*/*v*) ammonium sulfate. The solution was stirred over night at 4 °C. Precipitated proteins were centrifuged (4500 rpm, 30 min, 4 °C), washed once in 20 mL 0.1 M MOPS buffer (0.1 M MOPS, 0.05 M NaCl, 0.01 M CaCl_2_, pH 7.0) with 60% (*v*/*v*) ammonium sulfate, and centrifuged again (4500 rpm, 20 min, 4 °C). All proteins were stored in a total volume of 5 mL 0.1 M MOPS buffer with 60% (*v*/*v*) ammonium sulfate at 4 °C. For renaturation, 1 mL of the solution with the precipitated proteins was centrifuged (3 min, 13,000 rpm), and the pellet was dissolved in 500 μL buffer (1× MOPS, pH 7.0) over night on ice. The protein concentration was determined by A_280_ measurement by a spectrophotometer (Eppendorf, Hamburg, Germany). Enzymatic activities on barley-β-glucan (BBG), phosphoric acid swollen cellulose (PASC), and cellulose powder MN301 were measured by liberation of reducing sugars [23], as described previously [22]. In brief, all protein-substrate combinations were incubated as biological duplicates in reactions containing 100 μL 1% (*w/v*) substrate (BBG, MN301, PASC), 15 μL buffer (1× MOPS, pH 7.0), 0.1 μg × μL^−1^ proteins, and filled up to a final reaction volume of 200 μL with MQ H_2_O. The reaction tubes were incubated in a water bath at 55 °C for 30 h. The enzymatic digestion was stopped on ice and remaining substrate was removed by centrifugation. Fifty microliters digest samples or glucose standard samples (0.1–2.0 mg·mL^−1^) were mixed as technical triplicates with 75 μL DNSA solution (10 g·L^−1^ (*w/v*) 3,5-Dinitrosalicylic acid (DNSA), 10 g·L^−1^ (*w/v*) NaOH, 200 g·L^−1^ (*w/v*) sodium-potassium tartrate, 0.5 g·L^−1^ (*w/v*) sodium sulfate, and 2.0 g·L^−1^ (*w/v*) phenol) in a deep 96-well PCR plate. Plates were incubated at 95 °C in a thermoblock for 5 min and cooled on ice. The absorption of 100 μL solution was measured in a flat-bottom 96-well plate at 540 nm with a Sunrise^TM^ photometer (Tecan Group Ltd., Männedorf, Switzerland).

### 2.8. Species Identification, Sequencing, and Annotation of D. raffinosedens, Isolate 249c-K6

For species identification of the isolates obtained after cultivation, genomic DNA was extracted from 1 mL corresponding bacterial culture using the Soil DNA Extraction Kit (Roboklon, Berlin, Germany, manufacturer’s instructions). The 16S rRNA gene was amplified and analyzed as described previously [20]. Presence of *H. thermocellum* was verified via species-specific primers (CthF: TTTACCGGAAGTATATCCTAG and CthR: AATCATCTGCCCCACCTTC), as described previously [24].

For extraction of *D. raffinosedens* 249c-K6 high molecular weight genomic DNA, a fresh overnight culture, grown in GS2 medium at 55 °C, was pelleted (10 min, 5000 rpm, 8 °C) and washed twice in 5 mL 0.9% (*w/v*) NaCl solution. The pellet was resuspended in 5 mL lysis buffer (20 mM Tris/HCl pH 7.5; 25 mM EDTA; 75 mM NaCl, 1 mg × mL^−1^ lysozyme (AppliChem, Darmstadt, Germany)) and incubated for 1 h in a water bath at 37 °C. After adding 500 µL 10% (*w/v*) SDS, 500 µL proteinase K (10 mg × mL^−1^; AppliChem, Darmstadt, Germany) and 50 µL RNase A (AppliChem, Darmstadt, Germany; DNase free), the solution was incubated at slow agitation for 2 h at RT. Then, 2 mL NaCl (5 M) and 3 mL chloroform/isoamyl alcohol (Roth) were added followed by incubation with slow agitation at RT for 30 min. After centrifugation (10 min, 5000 rpm, 20 °C), the upper liquid phase was transferred into a fresh falcon tube. After addition of 3 mL chloroform/isoamyl alcohol (Carl Roth, Karlsruhe, Germany), the solution was incubated for another 30 min with slow agitation at RT and centrifuged (10 min, 5000 rpm, 20 °C). The upper liquid phase was transferred into a fresh falcon tube. These three steps were repeated until a clear phase separation was visible. The upper liquid phase was transferred again into a fresh falcon tube. Subsequently, DNA was precipitated by adding one volume of 100% (*v*/*v*) isopropanol (RT) and gentle agitation. Under a PCR hood, the high molecular weight DNA was grasped with a pipette tip and immersed two times in 500 µL cold (−20 °C) 70% (*v*/*v*) ethanol. The dried DNA was dissolved in 100 µL MQ H_2_O.

For genome sequencing of *D. raffinosedens* isolate 249c-K6 at the ZIEL—Institute for Food & Health, Core Facility Microbiome/NGS, Technical University of Munich, Freising, Germany, 1 µg chromosomal DNA of a clonal pure culture was used to prepare a DNA library using the TruSeq DNA PCR-free sample preparation kit (Illumina, Inc., San Diego, CA, USA). A protocol optimized for DNA shearing and fragment size selection was applied [25]. The library was sequenced using the Illumina HiSeq system with about 3.2 million 2 × 150 base pairs (bp) reads in paired-end (PE) mode, according to the manufacturer’s instructions. The raw sequencing data were assembled using SPAdes v. 3.11.1 [26]. Default parameters were used for all software unless specified otherwise.

The NCBI Prokaryotic Genome Annotation Pipeline [27] was used for prediction and annotation of open reading frames (ORFs) in the *D. raffinosedens* 249c-K6 genome. Subsequently, all automatically annotated protein sequences of the whole genome sequence were annotated to K numbers with the BlastKOALA tool [28] specifying family or genus of nearest related organism (taxid: 1185408). Metabolic pathways were reconstructed and analyzed via the Kyoto Encyclopedia of Genes and Genomes (KEGG) database webpage (https://www.genome.jp/kegg/) using the KEGG mapper tool [29]. Additionally, carbohydrate-active enzymes (CAZymes) were annotated with the dbCAN2 web server (National Science Foundation; http://bcb.unl.edu/dbCAN2/blast.php) [30] utilizing HMMER [31] and DIAMOND [32], including the prediction of signal peptides via SignalP 4.0 [33]. Enzymatic function of automatically predicted CAZymes of special interest, here glycoside hydrolases with signal peptide, were checked manually with the Carbohydrate Active Enzymes (CAZy) database (http://www.cazy.org/) [34].

For phylogenetic analysis on the whole-genome level, the draft genome was compared with a multiple alignment of 92 up-to-date bacterial core genes (UBCG) derived from 1492 species covering 28 phyla using the UBCG pipeline version 3 on the EZBioCloud webpage (https://www.ezbiocloud.net/tools/ubcg) [35]. Our application of UBCG, in essence, first extracts core genes from the genomes using Prodigal 2.6.3 and hmmsearch 3.1b2, aligns them using Mafft v. 7.310, and finally infers phylogenetic trees for each gene and the concatenated alignment of all genes using Fasttree v. 2.1.10. Reference genomes for phylogenetic analysis were chosen by performing a comparison of the 16S rRNA gene sequence from *D. raffinosedens* 249c-K6 with the nucleotide BLAST tool (BLASTn) against the Reference RNA sequences (refseq_rna) database of the NCBI (National Center for Biotechnology Information, http://www.ncbi.nlm.nih.gov) using the Megablast algorithm. From the top 20 hits, the whole genome sequences, if available at the NCBI, were downloaded and used for comparison. Further, the average nucleotide identity (ANI) was calculated via jspecies.org and the ANIb algorithm [36].

### 2.9. Prevalence and Activity of D. raffinosedens in Mesophilic and Thermophilic Biogas Fermenters Applying 16S rRNA Gene Amplicon Sequencing

To investigate the prevalence and activity of *D. raffinosedens* 249c-K6 in the four different reactor samples (m_st, m_ac, t_st, and t_ef), nucleic acids were extracted in parallel (two replicates, numbers 1 and 2, for each reactor type) using the GeneMATRIX Universal DNA/RNA/Protein Purification Kit (EurX, Gdansk, Poland). The sample was washed twice with 0.85% KCl [37] and 200 µL of each of the washed sample were mixed with 300 µL phenol and bead-beating in a Lysing Matrix E tube (MP Biomedicals, Eschewege Germany) for 2 × 20 s at level 5.0 in the FastPrep-24 Instrument (MP Biomedicals, Eschewege Germany). Two hundred microliters of Lyse All buffer (and 10 µL mercaptoethanol per mL buffer) and 300 µL of DRP buffer (and 10 µL mercaptoethanol per mL buffer) were added to 100 µL of the upper aqueous supernatant and transferred to a DNA binding spin-column and centrifuged. The flow-through was mixed with 300 µL ethanol (100% (*v*/*v*)) and applied to an RNA binding spin-column. Then, the extraction of RNA and DNA was performed according to the manufacturer’s protocol. The RNA was finally eluted in 60 µL RNase-free water and the DNA in 100 µL Elution Buffer. From 30 µL of the extracted RNA, possible DNA contamination was removed using the Turbo DNA-free Kit (Thermo Fisher Scientific, Waltham, MA, USA). The DNA-free RNA (5 µL) was reverse transcribed into cDNA using Multi-Temp Affinity Script Reverse Transcriptase (Agilent) and the 16S rRNA gene-specific reverse primer R1378 (5’–3′ reverse primer CGGTGTGTACAAGGCCCGGGAACG). Adapter sequences, TCGTCGGCAGCGTCAGATGTGTATAAGAGACAG (forward) and GTCTCGTGGGCTCGGAGATGTGTATAAGAGACAG (reverse), were attached to the gene specific primers F939 (5’–3’ forward primer GAATTGACGGGGGCCCGCACAAG) and R1378 targeting the 16S rRNA gene region V6–V8 of Bacteria [38]. With these modified primers, PCR was performed using Platinum Taq Polymerase (Thermo Fisher Scientific, Waltham, MA, USA) and running 25 cycles for amplification. After checking the amplicons on an agarose gel for the correct length they were sent to the ZIEL (Institute for Food & Health, Core Facility Microbiome at the Technical University of Munich) for further processing and sequencing via “paired-end sequencing by synthesis” on an Illumina MiSeq system with a sequencing length of 2 × 300 bp PE, following the manufacturer’s instructions.

Processing of raw MiSeq forward and reverse PE reads was done as described by Engel et al. [39] with the following minor adjustments to individual bioinformatic steps. To achieve a higher assembly rate, we assembled Miseq PE reads in an iterative manner using Flash v. 1.2.11 [40]. All reads failing the first round of read assembly were clipped to a q20 average quality threshold using sickle v. 1.33 [41] and re-submitted to flash. This process was repeated consecutively while increasing the quality clipping threshold by 3 up to the point where either all reads could be assembled or a maximum quality clipping threshold of q35 was reached. All other steps, i.e. adapter clipping with cutadapt v1.18 [42]; de-replication, alignment, filtering, and de-noising with mothur v. 1.41.3 [43]; chimera checking and OTU clustering with USEARCH v. 8.0.1477 [44]; and taxonomic classification on genus level based on a naïve Bayesian classifier with a 80% confidence cutoff and the full SILVA database v. 132 [45], were carried out as described in detail previously [39] but without applying a length filtering step after primer clipping. Afterwards, consensus sequences of resulting OTUs were aligned with the structural rRNA aligner SSU-ALIGN v. 0.1.1 [46] and the alignment was used for reconstructing the best maximum-likelihood phylogeny from 100 rapid bootstrap replicates using the RAxML v. 8.0 [47] with the GTRCAT model. Subsequently, statistical analysis of the 16S rDNA and reverse transcribed 16S rRNA amplicon sequencing was performed via RHEA [48] in R.

### 2.10. Occurrence of D. raffinosedens Sequences in Biogas-Producing Microbial Communities Deduced from 16S rDNA and Reverse Transcribed 16S rRNA Amplicons of this Study or from Publicly Available Metagenome Data

For a targeted identification of the organisms of interest, namely *D. raffinosedens* 249c-K6, in the DNA and cDNA amplicon reads, PE assembled and primer clipped reads were mapped with the aln and samse module of the BWA aligner v. 0.7.13 [49] against the EzBioCloud database v. 2018.05 [50]. Mappings were filtered to retain mapped reads with a minimum mapping length of 300 bases and a minimum percent identity of 99% to the reference (full length 16S rRNA gene sequence of *D. raffinosedens* 249c-K6 accession no MT350287) based on the CIGAR, NM, and XM tag of the resulting SAM file using SAMtools v. 1.8 [51] and custom scripts.

Additionally, publicly available metagenomic and metatranscriptomic datasets were analyzed for sequence identity to evaluate the occurrence of the *D. raffinosedens* 249c-K6 in biogas plants in general. The assembled genome of *D. raffinosedens* 249c-K6 was uploaded to the Microbial Genomes Atlas (http://microbial-genomes.org) [52] and compared to the project “biogas microbiome” described by Campanaro et al. [53] with default settings.

## 3. Results and Discussion

### 3.1. Isolation of D. raffinosedens from Cellulolytic Mixed Cultures Isolated from Biogas Fermenters

The overall aim of this study was to study and characterize the synergism of a hydrolytic/cellulolytic bacterial consortium isolated from biogas fermenters. Therefore, cellulolytic mixed cultures from a previous study [20] were examined in more detail. In this study, *H. thermocellum* was isolated repeatedly from these cultures (data not shown). One representative isolate, *H. thermocellum* strain 250c-K4, was further analyzed. Interestingly, from mixed cultures dominated by *Defluviitalea* and *Hungateiclostridium* spp. [20], the separation of *Defluviitalea* spp. in pure, cellulolytic cultures was not possible by standard methods. These attempts resulted again in co-cultures dominated by *D. raffinosedens* and, in minor amounts, *H. thermocellum*, as verified by 16S rRNA gene sequencing and species-specific PCR, respectively (data not shown). The frequency of isolating *D. raffinosedens* growing in co-culture with *H. thermocellum* on crystalline cellulose also pointed to a putative synergism of these two organisms. As a representative for a co-culture of *D. raffinosedens* and *H. thermocellum*, the consortium 253c-K6 was further analyzed. 

Pure cultures of *Defluviitalea* spp. were obtained by changing the carbon source from crystalline cellulose to xylose and carrying out several transfers and cultivation in liquid medium before isolating single colonies from streak plates and re-inoculation of single colonies in liquid media. This approach was successful apparently because the saccharolytic species *D. raffinosedens* is able to use different mono- and disaccharides including pentoses such as xylose as carbon source [54], whereas *H. thermocellum* is not able to use pentoses for growth [55]. Subsequently, the clonal pure cultures were identified as belonging to *D. raffinosedens* showing 99.71% 16S rRNA gene sequence identity to *D. raffinosedens*, A6^T^ [54] and 96.29% identity to *D. saccharophila*, LIND6LT2^T^ [56]. The top 20 list showing the comparison of the 16S rRNA gene sequence of the isolate *D. raffinosedens* against the NCBI database are summarized in Appendix A. As a representative isolate, strain 249c-K6 was further analyzed and its 16S rRNA gene sequence was deposited at NCBI database under the accession number MN744427.

### 3.2. Genome Sequence Analysis of the Strain D. raffinosedens 249c-K6

To investigate the effect of the *H. thermocellum* cellulolytic activity in co-culture with *D. raffinosedens*, the genome of the strain *D. raffinosedens* 249c-K6 was sequenced and analyzed in detail. This is the first genome of the species *D. raffinosedens* sequenced thus far. The assembly of the whole genome sequencing resulted in 31 contigs (≥1000 bp) with a total length of 3,092,142 bp and an average 70-fold sequencing coverage. The average G + C content was 35.67 mol%. Gene prediction and manual annotation of the *D. raffinosedens* 249c-K6 genome sequence were performed applying the NCBI Prokaryotic Genome Annotation Pipeline [27] and resulted in the identification of 2,950 genes in total, thereof 2806 protein coding and 58 pseudo genes. Further, 86 RNA genes encoding 3 5S rRNAs, 7 16S rRNAs, 5 23S rRNAs, 67 tRNAs, and 44 non-coding RNAs were predicted. The genome sequence of *D. raffinosedens* 249c-K6 was deposited at NCBI database under the accession number PRJNA591875.

#### 3.2.1. Phylogenetic Classification of *D. raffinosedens* 249c-K6 in Relation to Members of the Genus Defluviitalea

Among the *Defluviitalea* members previously described in the literature [54,56,57], only for the relative strain *D. phaphyphila* Alg1 [57] complete genome sequence information is publicly available. Strains *D. raffinosedens* 249c-K6 and *D. phaphyphila* Alg1 showed an ANI value of 75.61% with 40.84% of the 249c-K6 genome aligned to the genome of *D. phaphyphila*, indicating that strain 249c-K6 belongs to another species distinct of *D. phaphyphila* [58,59]. Thus, the genome sequence of strain 249c-K6 provided here helps to elucidate the genome diversity within this genus. To determine the phylogeny of *D. raffinosedens* 249c-K6 in relation to other completely sequenced closely related members, as determined by comparison of the 249c-K6 16S rRNA gene sequence (Accession No. MN744427) analysis against the NCBI database (refsequ_rna), 14 reference strains including *D. phaphyphila* Alg1 were used to build the phylogenetic tree. The reference strains studied, except *D. phaphyphila* Alg1, shared only 88–89% 16S rRNA gene sequence identity to strain 249c-K6 (Appendix A). The phylogenetic tree demonstrated close relatedness of *D. raffinosedens* 249c-K6 to *D. phaphyphila* Alg1 supporting affiliation of the strain 249c-K6 to the genus *Defluviitalea* (Figure 1). Further, the UBCG tree sustains the phylogenetic distance to the reference strains already observed at 16S rRNA gene sequence level.

#### 3.2.2. *D. raffinosedens* 249c-K6 Metabolic Pathways Predicted by Means of KEGG

Reconstruction of metabolic pathways utilizing the BlastKOALA [28] and KEGG mapper tool [29] revealed a total of 37 complete pathway modules annotated in the draft genome of *D. raffinosedens* 249c-K6 (Appendix A). These included seven modules from the central carbohydrate metabolism: Glycolysis (Embden-Meyerhof pathway) (M00001), Glycolysis (M00002), Gluconeogenesis (M00003), Pyruvate oxidation (M00307), Citrate cycle (first carbon oxidation) (M00010), Pentose phosphate pathway, non-oxidative (M00007), and PRPP biosynthesis (M00005). Moreover, experimental physiological characterization revealed growth of 249c-K6 on xylose as carbon source (data not shown), which is consistent with the strain description of *D. raffinosedens* [54]. The pentose phosphate pathway is important for xylose utilization, as recently reported for *Pseudoclostridium thermosuccinogenes* [60]. Therefore, the presence of genes for the complete non-oxidative pentose phosphate pathway in the draft genome supports the experimentally observed xylose utilizing property of *D. raffinosedens* 249c-K6.

#### 3.2.3. *D. raffinosedens* 249c-K6 Genes Predicted to be Involved in Carbohydrate Utilization

To gain a better insight into the saccharolytic potential of *D. raffinosedens*, the draft genome was analyzed for CAZyme-encoding genes with the dbCAN2 meta server [30] and revealed a total of 53 CAZymes including 32 glycoside hydrolases (GH), 12 glycosyl transferases (GT), 2 carbohydrate esterases (CE), and 7 multi-domain enzymes. Of all GHs, only for six a signal peptide (SP) was predicted. As GHs with SPs are of particular relevance for the extracellular degradation of complex plant biomass, they were examined manually for their enzymatic function in the CAZy database [34]. This revealed the presence of GH13 (subfamily 36), GH32 and GH43 (subfamilies 4 and 16), as well as carbohydrate-binding modules (CBM) 6 and 38. For one multi-domain protein [CBM38(44-189) + CBM38(221-367) + CBM38(409-527) + GH32(808-1121) + CBM38(1313-1448)], enzymatic activity on the fructose polysaccharide inulin might be possible. Enzymatic activities within the identified (sub-)families are summarized in Appendix A.

### 3.3. Physiological Comparison of H. thermocellum as Pure Culture and in Co-Culture with D. raffinosedens 

To determine cellulose hydrolysis properties of *H. thermocellum* as pure culture in comparison to the co-culture with *D. raffinosedens*, strains were analyzed in three different media supplemented with cellulose powder MN301 as sole carbon source. The determination of cellulose-hydrolyzing activities by the Anthrone assay revealed higher activities, manifested as a larger extent of cellulose degradation, for 253c-K6, a co-culture of *H. thermocellum* and *D. raffinosedens*, compared to two pure cultures of *H. thermocellum* (ATCC27405^T^ and 250c-K4), independent of the medium used for cultivation (Figure 2A). Especially in GS2 and GS2.5GR medium, 99% of MN301 was hydrolyzed by the co-culture within two days of cultivation at 55 °C. *H. thermocellum* 250c-K4 alone hydrolyzed only 84% MN301 in GS2 and 93% in GS2.5GR within the same incubation period. As expected, *D. raffinosedens* 249c-K6 alone did not grow on cellulose as the sole carbon source. Similarly, higher amounts of total volatile metabolites and higher concentrations of acetic acid were detected by GC analysis for the co-culture 253c-K6, as compared to *H. thermocellum*, strains ATCC27405^T^ or 250c-K5, alone (Figure 2B). In contrast, the specific activity of culture supernatant proteins on different β-glucans (soluble (BBG), crystalline (MN301), and amorphous (PASC)) did not differ much when *H. thermocellum* was in co-culture with *D. raffinosedens* (253c-K6) compared to *H. thermocellum* alone (strains ATCC27405^T^ or 250c-K5) (Figure 2C). All three cultures showed highest enzymatic activity on BBG (1.8 mU × mL^−1^) and lowest on MN301 (0.3–0.6 mU × mL^−1^). Thus, the co-culture of *D. raffinosedens* with *H. thermocellum* showed enhanced cellulose hydrolysis as well as metabolic activity. However, cellulolytic enzyme activities of culture supernatant proteins were not increased when *H. thermocellum* was co-cultivated with *D. raffinosedens*. This indicates that the observed enhanced cellulose hydrolysis of the co-culture is rather caused by an (metabolic) interaction of both organisms and not by extracellular enzymes of *D. raffinosedens* that might have cleaved crystalline cellulose. Besides, *D. raffinosedens* is described as a non-cellulolytic, but saccharolytic organism [54]. Furthermore, co-cultures of *H. thermocellum* with other, non-cellulolytic organisms have previously been found in the context of lignocellulose fermentation to ethanol [18]. For example, a co-culture of *H. hermocellum* and *Thermoanaerobacter thermohydrosulfuricus* was reported to ferment hemicellulose from Solka Floc, in contrast to both organisms as pure cultures [61]. The authors proposed that by a metabolic interaction of both organisms, xylobiose and xylose released by (hemi-)cellulase activity of *H. thermocellum* were fermented by *T. thermohydrosulfuricus* to ethanol. *H. thermocellum* breaks down cellobiose and cellodextrins intracellularly by phosphorolytic cleavage and utilizes only the formed glucose-1-phosphate, whereas glucose is liberated into the medium [62]. Thus, by growth on hemicellulose, pentose sugars liberated by *H. thermocellum* can be utilized via cross-feeding by other organisms. In the case of *H. thermocellum* in co-culture with *D. raffinosedens*, we suppose that the saccharolytic organism *D. raffinosedens* uses the liberated soluble sugars for its growth. These soluble sugars thereby are removed and cannot act as end-product inhibitors of *H. thermocellum* cellulases anymore, which plausibly explains the enhanced cellulose hydrolysis.

Cultures: ATCC27405^T^ and 250c-K4, *H. thermocellum* alone; 253c-K6, *H. thermocellum* and *D. raffinosedens* in co-culture. Co-cultures are highlighted with ‘#’.(A) All cultures were incubated as biological duplicates in three different media (GS2, GS2.5GR and GR20) with a total volume of 50 mL and 0.2% (*w/v*) MN301. Residual MN301 after cultivation was determined by Anthrone assay. Data were normalized to an incubation period of two days. *D. raffinosedens* alone did not grow on MN301 as sole carbon source and is therefore not included in the figure (residual substrate 100%).(B) Metabolite production was compared using three different cultivation media (GS2, GS2.5GR, and GR20) in biological duplicates. Data were normalized to an incubation period of two days at 55 °C. One hundred microliters culture supernatant were measured in a total volume of 500 μL with 50 μL of 0.5% (*v*/*v*) 1-propanol as internal standard for quantification.(C) All digests were performed as biological duplicates in a total volume of 200 μL with 0.5% (*w/v*) substrate in a water bath at 55 °C for 30 h. Protein concentration within the batch was 0.1 μg·μL^−1^. DNSA assay was performed in technical triplicates. Enzymatic activity was calculated with Equation (1):
$$specific\;activity\;\left(\frac{U}{ml}\right)=\frac{m_{glu}\times 1000}{M_{glu}\;\cdot \;V_{e}\;\cdot \;c_{e}\;\cdot \;t}$$
**Equation 1:** Calculation of the specific enzymatic activity via DNSA assay.
U is units (μmol × min^−1^); m_glu_ is the mass of glucose (μg) calculated by means of glucose standard curve; M_glu_ is the molar mass of glucose (μg × μmol^−1^); t is the incubation period (min); V_e_ is the volume of enzyme within the batch (µL); and c_e_ is the concentration of enzyme (μg × μL^−1^).

### 3.4. Importance of the Genus Defluviitalea in Microbial Communities of Biogas Fermenters

#### 3.4.1. Comparison of Microbial Community Members’ Abundance and Activity as Deduced by 16S rDNA and Reverse Transcribed 16S rRNA Amplicon Sequencing

Taxonomic composition and the active part of the microbial biogas communities originating from two thermophilic and mesophilic lab-scale maize-fed biogas reactors operated in parallel (numbered 1 and 2) were examined for the presence of *Defluviitalea* species in these communities. For this purpose, the hypervariable regions V6-V8 of the bacterial 16S rRNA gene were analyzed in detail by means of 16S rRNA (cDNA) and 16S rDNA (DNA) amplicon sequencing (Figure 3). Two mesophilic, stable (m_st) and acidified (m_ac), and two thermophilic, stable (t_st) and highly efficient (t_ef), process conditions were investigated. The corresponding sequence datasets of the 16S rRNA amplicon libraries of this study are available in the ENA repository under the Bioproject ID PRJEB37872.

Looking at DNA, the taxonomic composition at higher taxonomic ranks was highly similar between the fermenter replicates (data not shown). At phylum level, the microbiomes were mainly composed of *Firmicutes* (on average 74% in the thermophilic vs. 49% in the mesophilic processes), followed by *Bacteroidetes* (on average 13% in the thermophilic vs. 19% in the mesophilic processes). In the thermophilic biogas-producing communities, *Synergistetes* were also significant (on average 6%), while the mesophilic microbiomes showed high abundance of *Actinobateria* (5% and 18% in m_st and ma_ac, respectively) and, at stable process conditions only, *Spirochaetes* (on average 34% in m_st). 

Further down in the taxonomic hierarchy, a clear abundance reduction of the class *Clostridia* was observed under mesophilic conditions (31% and 36% in m_st and m_ac, respectively) in comparison to the thermophilic reactors (71% and 75% in t_st and t_ef, respectively) (Figure 3). Sequences of the genus *Defluviitalea*, a member of the class *Clostridia*, were detected in all thermophilic microbiomes, but less in the DNA samples t_st (on average 0.26%) than in t_ef (on average 0.52%). Based on DNA analysis, the thermophilic microbiomes were less diverse than the mesophilic microbiomes (e.g., Shannon indices on average 2.84 and 3.4 for thermophilic or mesophilic microbiomes, respectively; for details, see Appendix A). This is in line with the observed higher species diversity in mesophilic microbial biogas communities, as described previously [9,63,64].

The metabolically active part of the bacterial biogas communities, according to 16S rRNA transcript (cDNA) analysis, showed that mainly *Firmicutes* (58% in t_st and 65% in t_ef), *Synergistetes* (19% in t_st and 10% in t_ef), *Bacteroidetes* (11% in t_st and 18% in t_ef), and *Halanaerobiaeota* (11% in t_st and or 3% in t_ef,) were highly active at thermophilic conditions. At mesophilic conditions, *Firmicutes* (52% in m_st and 51% in m_ac), *Actinobacteria* (13% in m_st and 30% in m_ac), and *Bacteroides* (22% in m_st and 13% in m_ac) were transcriptionally most active. *Spirochaetes* were found to be highly abundant at mesophilic process condition (34% in m_st DNA) but showed relatively poor transcription (3% in m_st cDNA).

Analysis of the β-diversity (Figure 4) revealed significant dissimilarities between the different process conditions in general (*p* = 0.0001). The bacterial composition deduced from DNA and cDNA at the two mesophilic process conditions, m_st and m_ac, differed highly between each other as well as compared to the thermophilic process conditions. In contrast, the two thermophilic process conditions, t_st and t_ef, were closely related to each other.

As *Defluviitalea* spp. were of particular interest, transcriptional activity of this genus in the thermophilic and mesophilic microbiomes was examined in detail. The highest transcriptional activity of the genus *Defluviitalea* (1.44%) was observed in the microbiome originating from the thermophilic biogas reactor operating highly efficiently (reactor replicate 2). At mesophilic conditions, only a very small portion of cDNA sequences could be classified as belonging to the genus *Defluviitalea* (0.07% at maximum in m_ac reactor replicate 2; see also Appendix A). Furthermore, a targeted mapping approach for the organism of interest, namely *D. raffinosedens* was performed to verify the presence of the latter in the community. In total, 468 reads mapped with >99% identity to the reference gene [65,66] (full-length 16S rRNA gene; Accession No. MT350287). All hits were observed in DNA or cDNA of the thermophilic reactors only, and not at mesophilic processes (see also Appendix A). The abundance and transcriptional activity of this species were both higher in t_ef than in t_st, and the relation of the transcripts (cDNA, 321 hits in total) to the genes (DNA, 147 hits in total) is in line with the results on the broader genus level (Appendix A).

We observed a positive correlation between: (i) the abundance of the OTUs representing *Defluviitalea* spp., the exclusive presence of reads mapped to *D. raffinosedens* in the thermophilic processes, and a higher number of cDNA than DNA in either case in the sequence libraries; and (ii) the efficiency thermophilic biogas process. This emphasizes the role of this non-cellulolytic organism for thermophilic lignocellulose hydrolysis as already discussed in the previous section. Moreover, the Blastn analysis of the 16S rRNA gene sequence MN744427 against the NCBI nucleotide collection (nr/nt) revealed that of the 21 nucleotide sequences with identities above 98%, eight were related to “biogas” and seven to “cellulose” when analyzing the title of the deposited nucleotide sequence (Appendix A). This indicates that sequences belonging to the genus *Defluviitalea* have been detected earlier but could not be assigned to a prokaryote with standing in nomenclature. Thus, identifying *D. raffinosedens* in other publicly available datasets from biogas-related environments (see below) is of great interest.

#### 3.4.2. The Occurrence of Defluviitalea Genus in Biogas-Producing Plants as Deduced from Publicly Available Metagenome Data

A recently published study by Campanaro et al. [53] described the establishment of a repository on microbial genome sequence information, which is of great importance for future studies and broadens our understanding of how the microbes in biogas processes contribute to the anaerobic digestion process. The authors analyzed 134 publicly available metagenome datasets derived from different biogas reactors and recovered 1635 metagenome-assembled genomes (MAGs) representing different bacterial and archaeal species. The genus *Defluviitalea* was detected in almost all microbiomes analyzed, but differed in abundance values. The highest amount of *Defluviitalea* was found in the microbiome originating from the thermophilic (55 °C) laboratory-scaled packed-bed biogas reactor fed with acetate and yeast extract as the sole sources of carbon and energy [53,67]. In the analysis of the MAGs originating from the corresponding microbiome, the MAG METABAT AS04akNAM 106 was identified as belonging to the genus *Defluviitalea* (please refer to Additional File 8 of the study by Campanaro et al. [53]). The responsive MAG featured a completeness value of 95.5% and contamination value being 5.4%. Hence, an ANI analysis [58], which is suitable for species demarcation [59], was calculated between the MAG METABAT AS04akNAM 106 and *D. raffinosedens* 249c-K6. The analysis revealed an ANI value of 99.5% by comparing 96.7% of the entire MAG sequence, indicating that these two members belong to the same species. Based on the obtained results and the abundance values published previously for *Defluviitalea* members [53,68], Bacteria highly related to the reference strain *D. raffinosedens* 249c-K6 play an important role within the community of the thermophilic biogas plant.

## 4. Conclusions

In this study, metabolic interactions of *D. raffinosedens* and *H. thermocellum* in respect of microbial conversion of plant-based biomass to biogas were studied. We observed a high frequency of isolating *D. raffinosedens* in co-culture with *H. thermocellum* and saw higher cellulose hydrolysis as well as a higher production of volatile metabolites for this co-culture in comparison to *H. thermocellum* alone. The results obtained point to a cross-feeding process between *H. thermocellum* and *D. raffinosedens.* It is proposed that enzymatic activity of *H. thermocellum* liberates soluble mono- and oligosaccharides from cellulose and hemicellulose, thereby promoting growth of the non-cellulolytic, but saccharolytic *D. raffinosedens*. The results obtained clearly indicate that the activity of the latter in turn intensifies cellulose hydrolysis by preventing feedback-inhibition and improving the thermodynamics of the degradation process. Bacterial synergism as observed in this study is supposed to accelerate biomethane production in anaerobic digestion of plant fibers by increasing the overall cellulose hydrolysis rates and increasing the amounts of produced volatile metabolites in a given time. We describe the *D. raffinosedens* 249c-K6 genome, which is the first of this species and the second within this genus, and therefore provide the first reference sequence of this species. Future studies will certainly benefit from the comprehensive genomic information on *D. raffinosedens* 249c-K6, e.g. by integrating this knowledge into models describing interactions within corresponding complex communities.

## Figures and Tables

**Figure 1 microorganisms-08-00915-f001:**
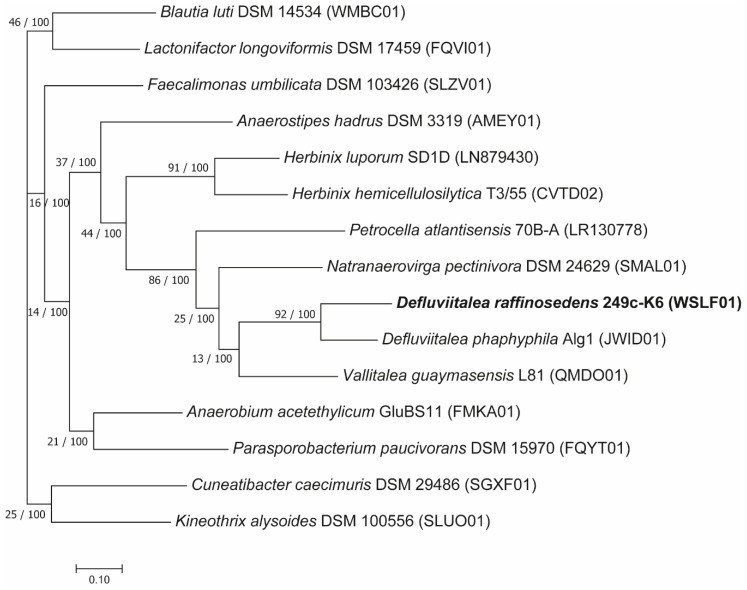
Phylogenetic position of *D. raffinosedens* isolate 249c-K6 in relation to the most closely related members of the class *Clostridia*. The tree was constructed applying the UBCG pipeline v. 3 [35] using standard settings based on 92 up-to-date bacterial core genes. In total, 86,940 nucleotide positions were used to build the concatenated alignment. For all branches, the number of examined genes (GSIs, gene support indices) supporting the given topology and bootstrap values are shown as follows: GSI/Bootstrap value. Bar, 0.01 substitutions per position.

**Figure 2 microorganisms-08-00915-f002:**
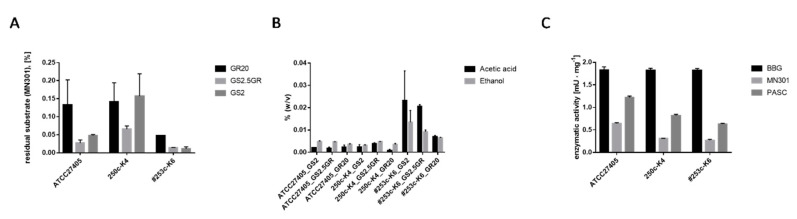
(**A**) Determination of cellulose hydrolysis for three different cultures with *H. thermocellum. (***B**) Detection of volatile metabolites from three different cultures with *H. thermocellum* via gas chromatography. (**C**) Specific activity (mU·mg^−1^) of supernatant proteins on different β-glucans, soluble (BBG), amorphous (PASC), and crystalline (MN301), from three different cultures with *H. thermocellum* determined by liberation of reducing sugars.

**Figure 3 microorganisms-08-00915-f003:**
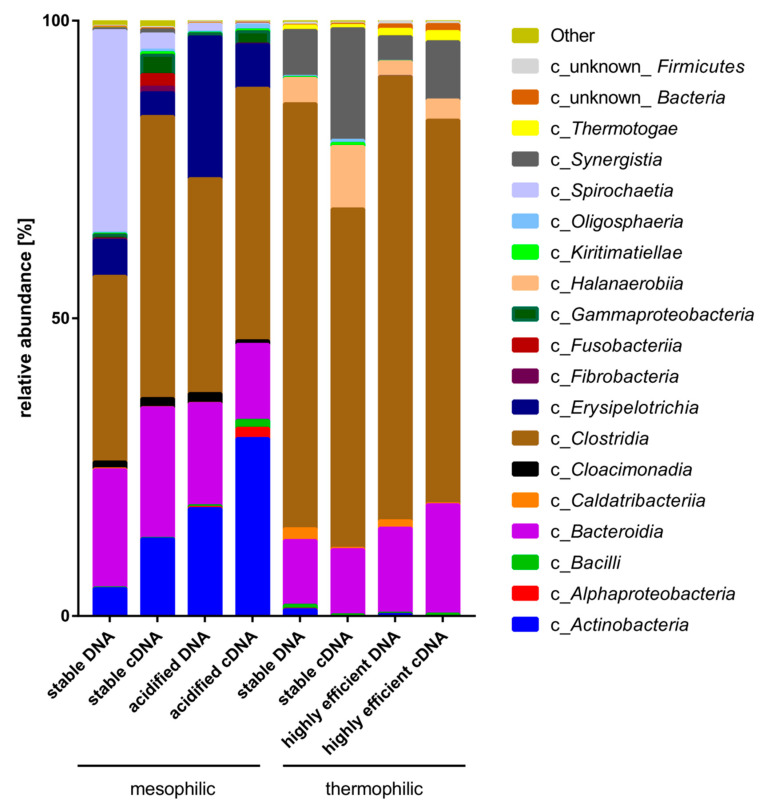
Taxonomic classification at class level of 16S rRNA (cDNA) and 16S rDNA (DNA) amplicon sequences originating from laboratory-scale of biogas reactors under four different process conditions. Values for the relative abundance at class level were averaged over two biological replicates.

**Figure 4 microorganisms-08-00915-f004:**
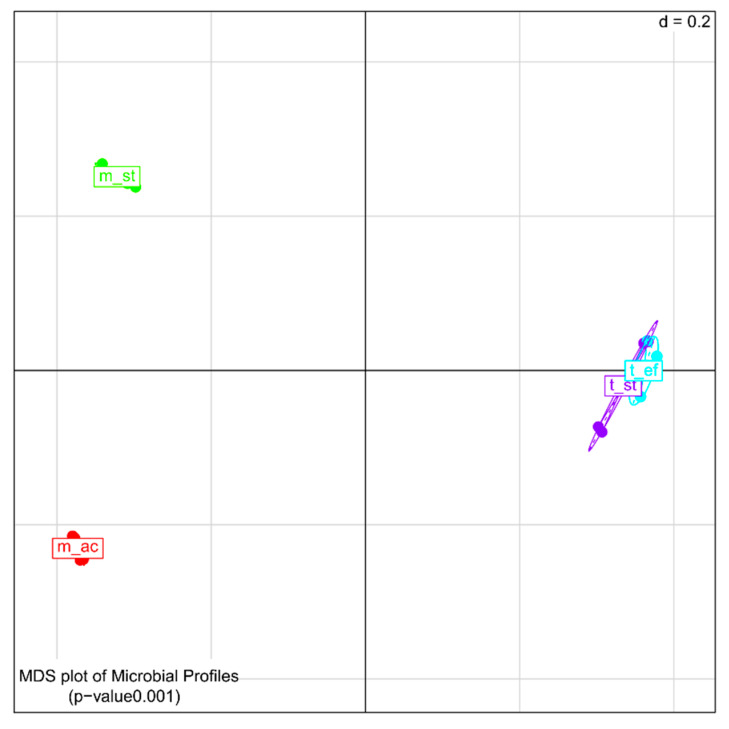
Multi-dimensional scattering (MDS) of the β–diversity of the microbial profiles derived from 16S rDNA and reverse transcribed 16S rRNA amplicon sequences of biogas reactors under four different process conditions with two biological replicates. Samples were grouped for process condition: m_st, mesophilic (38 °C), stable; m_ac, mesophilic, acidified; t_st, thermophilic (50 °C), stable; t_ef, thermophilic, highly efficient.

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
