# Peer review of "Importance of Defluviitalea raffinosedens for Hydrolytic Biomass Degradation in Co-Culture with Hungateiclostridium thermocellum"

_microorganisms, 2020, doi:10.3390/microorganisms8060915_

Round 1

Reviewer 1 Report

This manuscript targets at sequencing the genome of Defluviitalea raffinosedens and investigating its role during anaerobic digestion of biomass in co-culture with Hungateiclostridium thermocellum. In general, the manuscript is well motivated and of interest for the research in its field. However, the following comments, should be considered prior to its publication:

  • The major critical point of this manuscript is that it is very detailed which difficulties the reading. Authors should make an effort to place non-essential information as Supplementary files. This is specially required for the Methods section since some techniques such as software analyses can be placed as Appendix A. Furthermore, the main conclusions should be highlighted to identify the major scientific findings of the study.
  • Authors states that major conclusions are based on hypotheses which must be confirmed in subsequent works. Here, authors should provide more information about what the experiments needed to investigate and support the major claims from this study are.
  • Full microbial names are used the first time they appear (e.g. lines 57, 61, and 310) and abbreviations are used subsequently (e.g. lines 273 and 277). Furthermore, abbreviation “Hc. thermocellum” should be replaced by “H. thermocellum”.
  • What was the inoculum concentration in terms of total solids and volatile solids, and the substrate to inoculum ratio?
  • Line 288: replace “is not able use” by “is not able to use”.
  • Supporting information should be organized according to the order of appearance in the main text.
  • Citations should be removed from the conclusion section.

Author Response

Point-to-Point Response to Reviewer Reports

Manuscript ID: microorganisms-824223

We thank the Reviewers for the valuable comments, suggestions, and criticisms on our manuscript. We addressed all points of the Reviewer’s Report and recorded modifications introduced into the manuscript and responses to Reviewer comments in this Point-to-Point Response.

General comments of the Reviewer #1:

This manuscript targets at sequencing the genome of Defluviitalea raffinosedens and investigating its role during anaerobic digestion of biomass in co-culture with Hungateiclostridium thermocellum. In general, the manuscript is well motivated and of interest for the research in its field. However, the following comments, should be considered prior to its publication:

  • The major critical point of this manuscript is that it is very detailed which difficulties the reading. Authors should make an effort to place non-essential information as Supplementary files. This is specially required for the Methods section since some techniques such as software analyses can be placed as Appendix A.

Response: The authors are convinced that the detailed description of the work that has been done is important for the understanding and reproducibility of the results obtained. Also Reviewer 2 highlighted in particular that the manuscript was well written and appreciated the detailed description of the methods applied. In our opinion, this is the right way how to present the methods applied and the results obtained and to guide the reader through our study. However, to follow the criticism of the reviewer we have therefore included the additional information on the inocula, requested of Reviewer #1 (see below), as appendix in Supporting Information Table 1.

  • Furthermore, the main conclusions should be highlighted to identify the major scientific findings of the study.

Response: We revised our conclusions to be more precisely. The following sentences were added to the Conclusion section (see line 513 and 552) as suggested by the Reviewer#1: “We observed a high frequency of isolating D. raffinosedens in co-culture with H. thermocellum and saw higher cellulose hydrolysis as well as a higher production of volatile metabolites for this co-culture in comparison to H. thermocellum alone. [...] We describe the D. raffinosedens 249c-K6 genome, which is the first of this species and the second within this genus, and therefore provide the first reference sequence of this species.”

  • Authors states that major conclusions are based on hypotheses which must be confirmed in subsequent works. Here, authors should provide more information about what the experiments needed to investigate and support the major claims from this study are.

Response: The statement “However, a clear proof of this bacterial interaction should be elaborated in more detail in subsequent work” is a leftover from an earlier version of the manuscript. The authors (and also Reviewer#2 and #3) are convinced that the characterization of Defluviitalea raffinosedens has been carried out extensively which led to the constructive statements provided within the conclusions section. We removed the sentence mentioned earlier from the manuscript and withdraw the statement concerning subsequent work. We apologize for this inconvenience.

  • Full microbial names are used the first time they appear (e.g. lines 57, 61, and 310) and abbreviations are used subsequently (e.g. lines 273 and 277). Furthermore, abbreviation “Hc. thermocellum” should be replaced by “H. thermocellum”.

Response: We thank Reviewer #1 for this comment. We checked first time appearance of all microbial names and changed the abbreviation from Hc. thermocellum to H. thermocellum throughout the manuscript.

  • What was the inoculum concentration in terms of total solids and volatile solids, and the substrate to inoculum ratio?

Response: We considered the Reviewer’s criticism and provided further information on inoculation ratio for bacterial cultures in line 108. More detailed description on inocula used for the mesophilic and thermophilic biogas fermenters is given as Supporting Information Table 1 (see also Material and Methods section in line 75).

  • Line 288: replace “is not able use” by “is not able to use”.

Response: Line 288 was changed accordingly.

  • Supporting information should be organized according to the order of appearance in the main text.

Response: As recommended, supporting information was rearranged and the citations in the manuscript were adjusted accordingly.

  • Citations should be removed from the conclusion section.

Response: All citations were removed from the conclusion section by removing the part that needed citations (Line 545)

Reviewer 2 Report

I found the manuscript entitled “Importance of Defluviitalea raffinosedens for hydrolytic biomass degradation in co-culture with Hungateiclostridium thermocellum” very interesting and bringing important new details about optimal composition of microbiota for biogas production process. This paper brings new whole genome data (first Defluviitalea raffinosedens and second from these genus) and points out the role of this species as an accompanying organism of main cellulolytic organism, Hungateiclostridium thermocellum. Experiments and bioinformatic analysis done bring enough scientific bases for conclusions given by the Authors. The manuscript is very well written and contains detailed descriptions of all methods used, what is of great value. I have only one line of criticism (or rather suggestion) considering merits of the paper. It concerns phylogenetic analysis. Due to relatively new, not well recognized method used, based on a collection of core genes (UBCG (up-to-date bacterial core gene) I would suggest more precise description of components of a pipeline applied in this method (e.g. MAFFT for multiple sequence alignment and RAxML and FastTree for tree building are well-known and widely recognized algorithms, opposite to UBCG). Furthermore, I found showing number of genes supporting particular branch near tree vertices rather confusing as usually bootstrap results are shown in this way, therefore at quick glance a reader may have an impression that cladogram shown is highly unstable one.

Other my remarks consider results presentation and editorial aspects.

I was surprised that results of volatile metabolites production and amylolytic activity assay are presented as supplementary material. In my opinion both should be included in main body of the article, as two panels of one figure perhaps, or together with existing Figure 2.

All three above mentioned figures (Fig2 and Supplementary Fig 1 and 2) should be improved by change of culture symbols – exact number of the collection strain should be explained in caption but on the graph it would be better to see easily which culture is a co-culture. The symbol "253c-K6" (co-culture) is analogous in its syntax to 250c-K4 which is assigned to Hc. thermocellum alone, which I found very confusing.

Another issue is a title “enzymatic activity” of an y axis in supplementary Figure 2, and the same term used in the main body text, line 368. This statement is too general and most probably should be changed into "amylolytic activity" (because DNSA test was used here).

The sentence in lines 402-3 has no predicate. Probably it was meant to be a title of a paragraph? – Maybe bolding should be used?

The sentence in line 435 – probably should belong to Figure 3 caption but looks like a paragraph title.

The sentence in lines 481-2 - probably was meant to be a title of a paragraph – maybe bolding should be used?

I couldn’t find "MN301" in the list of abbreviations.

When these minor issues will be addressed by the Authors I have no doubts to strongly recommend this manuscript for publication.

Author Response

 Point-to-Point Response to Reviewer Reports

Manuscript ID: microorganisms-824223

We thank the Reviewers for the valuable comments, suggestions, and criticisms on our manuscript. We addressed all points of the Reviewer’s Report and recorded modifications introduced into the manuscript and responses to Reviewer comments in this Point-to-Point Response.

General comments of the Reviewer #2

I found the manuscript entitled “Importance of Defluviitalea raffinosedens for hydrolytic biomass degradation in co-culture with Hungateiclostridium thermocellum” very interesting and bringing important new details about optimal composition of microbiota for biogas production process. This paper brings new whole genome data (first Defluviitalea raffinosedens and second from these genus) and points out the role of this species as an accompanying organism of main cellulolytic organism, Hungateiclostridium thermocellum. Experiments and bioinformatic analysis done bring enough scientific bases for conclusions given by the Authors. The manuscript is very well written and contains detailed descriptions of all methods used, what is of great value.

  • I have only one line of criticism (or rather suggestion) considering merits of the paper. It concerns phylogenetic analysis. Due to relatively new, not well recognized method used, based on a collection of coere genes (UBCG (up-to-date bacterial core gene) I would suggest more precise description of components of a pipeline applied in this method (e.g. MAFFT for multiple sequence alignment and RAxML and FastTree for tree building are well-known and widely recognized algorithms, opposite to UBCG). Furthermore, I found showing number of genes supporting particular branch near tree vertices rather confusing as usually bootstrap results are shown in this way, therefore at quick glance a reader may have an impression that cladogram shown is highly unstable one.

Response:

We agree with the Reviewer #2 providing the statement that the UBCG pipeline is comparatively new, and therefore, should be introduced in much more detail. The current UBCG set was calculated using complete genomes of 1,492 microbial species covering 28 phyla, consisting of 92 genes published by Na et. al in 2018. This tool is available at ezbiocloud.net and allows the identification of core genes applying the Prodigal and Hmmsearch as well as 92 multiple alignments of these single-copy maker genes as well as the phylogenetic analysis of these alignments applying RAxML and FastTree.

Despite its relative novelty (since 2018), the UBCG pipeline is used within the field of phylogenetic analysis of (novel) procaryotic species and was cited 33 times (state 02.06.2020) in different scientific Journals. Examples from 2020 are:

  • Fontiers Microbiology (3389/fmicb.2020.00698)
  • Journal of Microbiology (1007/s12275-020-9294-1 & 10.1007/s12275-020-9376-0)
  • Pathogens (3390/pathogens9030204)
  • Applied & Environmental Microbiology (1128/AEM.02549-19)

Based on this information, we are convinced that UBCG tool is well-established and represents appropriate method for our analysis.

To follow the Reviewer’s criticism we now introduced the UBCG method in the corresponding Materia and Method chapter in line 206: “For phylogenetic analysis on the whole-genome level, the draft genome was compared with a multiple alignment of 92 up-to-date bacterial core genes (UBCG) derived from 1,492 species covering 28 phyla using the UBCG pipeline version 3 on the EZBioCloud webpage (https://www.ezbiocloud.net/tools/ubcg) [42]. Our application of UBCG, in essence, first extracts core genes from the genomes using Prodigal 2.6.3 and hmmsearch 3.1b2, aligns them using Mafft 7.310 and finally infers phylogenetic trees for each gene and the concatenated alignment of all genes using Fasttree v.2.1.10.”

            Further, the numbers at branch sites was changed to Gene Support Indices (GSI) and Bootstrap Values accordingly.

Other my remarks consider results presentation and editorial aspects.

  • I was surprised that results of volatile metabolites production and amylolytic activity assay are presented as supplementary material. In my opinion both should be included in main body of the article, as two panels of one figure perhaps, or together with existing Figure 2.

Response: We thank Reviewer#2 for this remark. We took his observation in account and changed Figure 2 by providing the panels A, B and C for presentation of the results concerning volatile metabolites production and enzyme activity assays (please see new Figure 2).

  • All three above mentioned figures (Fig2 and Supplementary Fig 1 and 2) should be improved by change of culture symbols – exact number of the collection strain should be explained in caption but on the graph it would be better to see easily which culture is a co-culture. The symbol "253c-K6" (co-culture) is analogous in its syntax to 250c-K4 which is assigned to thermocellum alone, which I found very confusing.

Response: Thank you for this remark. Co-culture are now highlighted with “#” in Figure 2 to simplify the differentiation.

  • Another issue is a title “enzymatic activity” of an y axis in supplementary Figure 2, and the same term used in the main body text, line 368. This statement is too general and most probably should be changed into "amylolytic activity" (because DNSA test was used here).

Response:     

The title of the y axis was not changed to simplify the graphic presentation. We now provided a precise description of the enzymatic acitivities measured via liberation of reducing sugars in the figure legend: “Specific activity (mU * mg-1) of supernatant proteins on different β-glucans, soluble (BBG), amorphous (PASC) and crystalline (MN301), from three different cultures with H. thermocellum determined by liberation of reducing sugars.“

Line 379 was changed accordingly: “In contrast, the specific activity of culture supernatant proteins on different β-glucans (soluble (BBG), crystalline (MN301) and amorphous (PASC)) did not differ much when H. thermocellum was in co-culture with D. raffinosedens (253c-K6) compared to H. thermocellum alone (strains ATCC27405T or 250c-K5).”

  • The sentence in lines 402-3 has no predicate. Probably it was meant to be a title of a paragraph? – Maybe bolding should be used?

Response: Was changed to bold (Line 432).

  • The sentence in line 435 – probably should belong to Figure 3 caption but looks like a paragraph title.

Response: Thank you for this remark. The sentence was integrated in the description of Figure 3.

  • The sentence in lines 481-2 - probably was meant to be a title of a paragraph – maybe bolding should be used?

Response: Was changed to bold (Line 509).

  • I couldn’t find "MN301" in the list of abbreviations.

Response: MN301 is a product name for crystalline cellulose powder MN301 from Machery Nagel. Line 106 was changed from ”cellulose powder MN301” to “crystalline cellulose powder (MN301; Machery Nagel)” for specification.

When these minor issues will be addressed by the Authors I have no doubts to strongly recommend this manuscript for publication.

Reviewer 3 Report

In this paper, metabolic interactions of D. raffinosedens and Hc. thermocellum in respect of  microbial conversion of plant-based biomass to biogas were studied. The results clarified to  a cross-feeding process between Hc. thermocellum and D. raffinosedens. The results shows enzymatic activity of Hc. thermocellum liberates soluble mono and oligosaccharides from cellulose and hemicellulose, thereby promoting growth of the non-cellulolytic, but saccharolytic D. The conversion of lignocellulosic plant  biomass to biogas by microorganisms is particularly inhibited by the complex fiber structure with  lignin incrustations of the substrate, protecting energy rich sugars from bacterial attack . I recommend this article for publication.

Author Response

We thank the Reviewer#3 for his positive evaluation of our article.

Round 2

Reviewer 1 Report

Authors have modified the manuscript according to major Reviewers' suggestions. Nevertheless, authors should upload the new version of supplementary information for revision prior to acceptance.